# Recent Advances in Theoretical Development of Thermal Atomic Layer Deposition: A Review

**DOI:** 10.3390/nano12050831

**Published:** 2022-03-01

**Authors:** Mina Shahmohammadi, Rajib Mukherjee, Cortino Sukotjo, Urmila M. Diwekar, Christos G. Takoudis

**Affiliations:** 1Department of Chemical Engineering, University of Illinois at Chicago, Chicago, IL 60607, USA; mshahm2@uic.edu; 2Vishwamitra Research Institute, Crystal Lake, IL 60012, USA; rmukhe0@gmail.com; 3Department of Chemical Engineering, University of Texas Permian Basin, Odessa, TX 79762, USA; 4Department of Restorative Dentistry, University of Illinois at Chicago, Chicago, IL 60612, USA; csukotjo@uic.edu; 5Department of Biomedical Engineering, University of Illinois at Chicago, Chicago, IL 60607, USA

**Keywords:** atomic layer deposition (ALD), precursors, mechanisms, deposition characteristics density functional theory, molecular dynamics, lattice Boltzmann method, Monte Carlo, group contribution method, computer-aided molecular design

## Abstract

Atomic layer deposition (ALD) is a vapor-phase deposition technique that has attracted increasing attention from both experimentalists and theoreticians in the last few decades. ALD is well-known to produce conformal, uniform, and pinhole-free thin films across the surface of substrates. Due to these advantages, ALD has found many engineering and biomedical applications. However, drawbacks of ALD should be considered. For example, the reaction mechanisms cannot be thoroughly understood through experiments. Moreover, ALD conditions such as materials, pulse and purge durations, and temperature should be optimized for every experiment. It is practically impossible to perform many experiments to find materials and deposition conditions that achieve a thin film with desired applications. Additionally, only existing materials can be tested experimentally, which are often expensive and hazardous, and their use should be minimized. To overcome ALD limitations, theoretical methods are beneficial and essential complements to experimental data. Recently, theoretical approaches have been reported to model, predict, and optimize different ALD aspects, such as materials, mechanisms, and deposition characteristics. Those methods can be validated using a different theoretical approach or a few knowledge-based experiments. This review focuses on recent computational advances in thermal ALD and discusses how theoretical methods can make experiments more efficient.

## 1. Introduction

Atomic layer deposition (ALD) is a vapor-phase technique to deposit thin-film materials on various substrates through sequential and self-limiting surface reactions [1]. ALD originated from two different methods: atomic layer epitaxy (ALE) and molecular layering (ML), which were first introduced in the 1970s [2]. ALD then emerged due to the need for precise film thickness on small devices with high aspect ratios [2,3,4]. The thin films produced by ALD are deposited using chemical gas or vapor phase species, called precursors, in a cyclic fashion [5]. Thus, ALD is often called an advanced form of chemical vapor deposition (CVD). However, unlike CVD, ALD consists of alternative pulses and purges of the precursors, resulting in deposition of the desired film with an expected thickness and composition [5,6]. Furthermore, ALD is often performed under vacuum and at various ranges of temperatures, from room temperatures up to high temperatures, due to which ALD has a wide temperature window [4].

Each cycle is composed of two half-cycles or half-reactions, in which up to one monolayer of a metal or metal oxide is deposited on the surface of the substrate [3]. During the first half-cycle, the precursor is carried by inert gas and pulsed into the reactor with a determined duration, reacts with the available active sites of the substrate, and is chemisorbed on the surface [5]. In this step, ideally, the precursor saturates the surface through a self-limiting reaction. Then, excess unreacted precursor molecules are purged out of the chamber using inert gas. In the second half-cycle, the co-reactant pulses into the reactor to react with the adsorbed precursor molecules on the substrate [7,8]. The common co-reactants for ALD are water vapor, O_2_, O_3_, and NH_3_ [9]. Finally, the excess co-reactant molecules and the by-products of the reaction are purged out of the reactor. These two half-cycles are repeated until the required film thickness and composition are achieved [10]. The schematic of an ALD cycle is presented in Figure 1.

The thin film deposition technique of ALD has a wide range of applications [3,5,9,10]. Some of the important applications of ALD include but are not limited to the following areas: semiconductor engineering [11,12,13], lithium-ion batteries [14,15,16], microelectromechanical systems (MEMS) [17,18], capacitors [19,20], fuel cells [21,22,23], solar cells [24,25,26], transistors [27,28,29], drug delivery [30,31], medical and biomedical fields [32,33,34,35], dental materials [6,10], and orthopedical implants [36,37,38]. Due to the capability of precisely depositing conformal ultra-thin films, ALD has found a wide range of applications in the fabrication of microelectronics such as gate oxides, semiconductors, and ferroelectrics [12,39,40]. This technique has also attracted much attention in medical and biomedical applications, where organic substrates such as polymers or biomaterials are required [6,34,41].

Previous works have reviewed the important advantages and disadvantages of ALD [5,10,42,43]. The films produced by ALD are ultra-thin with exact controllability over thickness, composition, and crystallinity at the Ångström level [4]. The films are also uniform, conformal, and pinhole-free across various substrates, even those with high aspect ratios or complex three-dimensional (3D) structures [43,44]. Additionally, many different metals, oxides, nitrides, sulfides, selenides, tellurides, fluorides, and metal–organic frameworks (MOFs) can be deposited by ALD as long as the associated precursors exist [3,9,10,42,45]. A wide range of temperatures can be used in ALD depending on different properties such as the nature of the materials involved and the final applications. ALD can produce high-quality films even at low substrate temperatures [46].

Being a slow process is one of the main drawbacks of ALD [3]. Depending on the conditions, a 100-nm thin film with thermal ALD would take around 20 h or more to deposit. Although spatial ALD (SALD) has been developed to overcome this drawback [47], it is a rather new tool and not always accessible. Moreover, SALD can be sensitive to ambient air if performed in open air, highly volatile precursors are needed for SALD, and, still, not many precursors for SALD exist [47]. Before each ALD experiment, all the conditions should be determined to find the optimized ones. Thus, for each ALD experiment, different questions need to be answered: What would be the final applications?What materials (i.e., precursor, oxidizer, and substrate) should be used?How long should the pulse and purge durations be to obtain a high-quality film?What would be the growth rate based on the determined conditions?What should be the temperature and pressure of the reactor and the precursor bubblers?Are all the materials stable in those conditions?

Deciding on each of these factors requires many experiments to adjust the conditions resulting in high-quality films and an optimized growth rate. Moreover, each experiment should be repeated a few times to validate the results and ensure the reproducibility of the samples. On the other hand, the common precursors for ALD are often expensive, and performing many experiments to tune the conditions is inefficient. Additionally, some of the mechanisms result in hazardous by-products, and it is preferred to reduce the number of experiments. Theoretical modeling is one of the ways to find optimal conditions without performing any experiments, which would tremendously reduce the cost and time associated with those experiments.

Recently, many research studies have used computer simulations and theoretical models in conjunction with ALD for a variety of purposes, such as the study of thin film structure and composition and material selection [48]. The modeling technique varies over multiple scales and is based on the interaction between atoms, molecules, particles, and groups of atoms, also known as functional groups. Here, we present an overview of the common theoretical methods used to model ALD and the insight that can be obtained about the process of ALD from the different types of modeling. Thus, we adopt a reverse engineering approach, where we define various aspects of ALD, describe the experimental procedure used for ALD process development, and describe how modeling and simulation have been used for a proper understanding and improvement of the ALD process. The rest of the paper is organized as follows. First, different aspects of ALD are briefly introduced and discussed, then the computational methods commonly used for modeling and simulation of ALD are presented, and, finally, recent studies on theoretical approaches to ALD are reviewed.

## 2. Aspects of ALD 

Extensive experimental and theoretical research has so far been performed on different aspects of ALD. Experimentalists often focused on one or some of the following areas: precursors, mechanisms happening in an ALD reactor, and deposition characteristics such as temperature window, saturating pulse and purge times, growth rate, composition, morphology, and surface properties of the deposited film. Since conventional ALD is a slow process and due to the disadvantages of SALD mentioned above, many time-consuming and expensive experiments are required to tune an ALD condition for desired applications. Thus, not many of the areas mentioned above would be examined in a single research study, and the researchers usually focus on a few of them when studying ALD systems.

### 2.1. ALD Precursors

An ALD precursor is often a metal surrounded by organic functional groups held in a vessel known as a bubbler. The bubbler temperature in an ALD system varies depending on the properties of the precursor. ALD precursors are volatile, thermally stable, and highly reactive [49]. A few physical properties are considered when selecting a precursor for an ALD study, such as materials of interest, reactivity toward the other co-reactant, ALD conditions, final applications, and desired film properties (e.g., dielectric constant, adsorption capacity, gas impermeability, leakage current, electrical conductivity, photochemical activity, and antimicrobial activity) [49,50,51,52,53].

Precursors should be able to react quickly with the active sites of the substrates and other precursor molecules. Hence, precursor chemistry is a key factor in an ALD process, which also affects the growth mechanisms. In recent years, the introduction of new precursors has attracted much attention. Theoretical methods are appropriate for designing new materials based on existing experimental data and predicting their properties before synthesizing them [6]. Moreover, first-principles calculations can be used to suggest new materials with predicted properties that would be proved later through experiments [39,54,55]. In addition, there is often the need to compare ALD precursors in terms of various aspects that could become practically impossible through experimentation. Theoretical approaches are useful for these purposes.

### 2.2. Deposition Characteristics

#### 2.2.1. Growth

The growth rate in ALD is defined as the thickness of the film divided by the number of cycles (nm/cycle) and depends on multiple factors, including the precursor flux reaching the substrate [5]. The growth rate is a prime factor in ALD studies since it provides a direct method to understand the amount of time each deposition takes and predict the film thickness before deposition. Thus, the growth rate is often reported for each system under study. In addition, the growth rate is affected by the reactor temperature and pressure, pulse and purge times of the reactants, and the nature of the substrate. The growth rate is often reported as a constant value (the slope of the plot showing film thickness versus the number of cycles); however, some studies reported that the growth rate might change over the number of cycles. For instance, a recent study reported duo-linear plots of TiO_2_ film thickness versus the number of ALD cycles. According to them, absorbed water molecules in pores of a polymethyl methacrylate (PMMA) substrate were released into the reactor during the initial cycles, affecting the growth mechanism and resulting in a higher growth rate compared with later deposition cycles (Figure 2) [6].

Another aspect of ALD growth is the growth mode, meaning how the materials are arranged on the surface of the substrate during ALD growth [56]. The ALD growth modes are mainly attributed to one of the following modes: Volmer–Weber growth, Frank–van der Merwe growth, and Stranski–Krastanov growth [57,58]. In the Volmer–Weber growth mode, also known as island growth, small clusters or islands are first nucleated on the surface. The reason for island nucleation, in this case, is that the interaction of the adsorbed atoms within themselves is stronger compared with the interaction between the atoms and the substrate. That is, the cohesive force within the atoms is stronger than the surface adhesive force, so the atoms tend to accumulate. Then, those small clusters grow into larger three-dimensional ones and reach each other, covering the whole surface [57].

On the other hand, in the Frank–van der Merwe growth mode, also known as bi-dimensional growth, the surface adhesive force is stronger than the intra atom cohesive force, leading to layer-by-layer growth on the substrate. That is, the atoms completely cover the surface, producing a complete monolayer before the subsequent layer is formed on top [59]. If both the Volmer–Weber and Frank–van der Merwe growth modes are combined, the growth mode is called Stranski–Krastanov. In the Stranski–Krastanov growth mode, which is the more common growth mode, the film starts to form on the surface of the substrate as a whole layer. After forming a rather thick film, the growth mode switches to island growth [58]. Where the transition happens is affected by the chemical and physical properties of both film and substrate materials [57,58,59]. Figure 3 presents a schematic of the three growth modes.

#### 2.2.2. Surface Morphology

Surface morphology is another characteristic of thin films usually examined and reported in ALD-related studies. Different techniques, such as atomic force microscopy (AFM), profilometry, and scanning electron microscopy (SEM), are used to scan the surface and provide three-dimensional images of surface topography. Surface morphology depends on many factors, including nucleation and growth mechanisms, growth rate, crystallinity, surface roughness, deposition conditions, substrate, and impurities [60,61].

#### 2.2.3. Surface Roughness

Another thin-film property that is often investigated in ALD studies is surface roughness. Roughness is a part of surface texture, defined as a measure of waviness or irregularities on the film surface. Mapping technologies such as AFM and optical or contact profilometry can be used to calculate the surface roughness. The surface roughness is significantly affected by the growth mode discussed earlier. For instance, the Volmer–Weber growth mode starts with island nucleation; the islands enlarge over time, and, most likely, their heights become larger than the thickness of one monolayer before they converge. Thus, if the growth follows the Volmer–Weber growth mode, the film is often made of rough layers. However, due to the layer-by-layer nature of the Frank–van der Merwe growth mode, smoother films are produced with this growth mode [57]. Aside from the growth modes, the surface roughness depends on other factors such as crystallinity, film thickness, and the nature of the substrate.

#### 2.2.4. Step Coverage (Conformality) 

Conformality can be defined as the deposition of a film with the same thickness on all topographic features, including the top, sides, and bottom surfaces of a three-dimensional substrate [62]. Since deposition in ALD happens through surface-controlled reactions, the thin films formed have excellent conformality. That is, the step coverage provided by ALD to the surface of complex structures is higher than that of conventional deposition processes such as CVD and physical vapor deposition (PVD). Conformality is a significant aspect, especially when there is a great desire to coat a complex three-dimensional nanostructured surface or substrates with high aspect ratios [63]. Recently, Cremers et al. [63] provided an extensive review of different aspects of our current knowledge about conformality in ALD processes.

#### 2.2.5. Deposition Temperature 

In thermal ALD, the temperature is the main driving force for the process. One of the terms in ALD studies is the temperature window, defined as the temperature range over which the self-controlled growth would occur at a constant rate. Inside the ALD window, the growth rate would change significantly with an increase in the temperature due to the physisorption/condensation of precursors on the surface or low reaction rates, leading to uncontrolled growth. At temperatures outside the ALD window, the precursor molecules may decompose or desorb the heated substrate, resulting in uncontrolled growth [64]. The typical temperature window for thermal ALD processes is 150–350 °C [5], although some studies reported a temperature window out of this range. For instance, for the organic and heat-sensitive substrates, the reactor temperature should be adjusted accordingly to prevent the deformation and degradation of the substrate. Room temperature ALD has been reported on collagen materials for biomaterial functionalization [34,65]. 

### 2.3. Thermal ALD Mechanisms

#### 2.3.1. Mechanisms

As previously mentioned, a binary ALD process consists of a dose–purge–dose–purge sequence of each reactant forming an ALD cycle. In thermal ALD, the surface reactions typically happen due to a relatively high temperature. The self-limiting nature of ALD allows for saturation to occur on the surface of the substrate during the dosage steps before the extra unreacted materials are purged out of the reactor. That is, after the chemisorbed species saturate all the available active sites, no more chemisorption would happen beyond that point, although more reactants exist in the reactor [66]. However, the exact mechanisms of ALD reactions remain sophisticated and challenging subjects. Nevertheless, some studies have modeled and predicted ALD reaction mechanisms. The ALD mechanisms are divided into three categories (initial surface reactions, reaction pathways, and precursor decomposition), each of which is briefly discussed below.

#### 2.3.2. Initial Surface Reactions 

A finite number of active sites are available on the surface of a substrate per each ALD cycle. Through the initial surface reactions in an ALD process, those active sites are occupied by the reactants and depleted at the end of each half-cycle, and then more active sites are created in a subsequent halfcycle. The initial growth per cycle in ALD depends on the number of nucleation sites and is categorized into three groups: linear, surface-enhanced, and surface-inhibited [67].

#### 2.3.3. Reaction Pathways

When describing the reaction pathways occurring in an ALD reactor, the formation of aluminum oxide (Al_2_O_3_) from trimethylaluminum (TMA) and water (H_2_O) is well-discussed. TMA acts as the aluminum precursor in that system, and water is the oxygen source. The simplified forms of some of the common and well-known reaction pathways in ALD processes are summarized and tabulated here (Table 1).

#### 2.3.4. Precursor Chemisorption

Previous studies are focused on describing different mechanisms of precursor chemisorption during an ALD process. These mechanisms are mainly categorized into three groups (ligand exchange, dissociation, and association) [66,72]. In ligand exchange chemisorption, the split of the precursor occurs on the surface, where its ligand is exchanged with a surface group, and a gaseous product is released [66]. In the dissociation process, one or more ligands of the precursors are separated from the molecule, bounded with the surface groups, and create active sites on the substrate [66]. The molecule fragments are usually separated due to an external source such as light or heat [72]. On the other hand, in the association process, no ligand is split from the precursor, and a coordinated bond is formed between the precursor and the surface active sites [66]. A schematic of these mechanisms was reprinted from reference [73] and is shown in Figure 4.

An example of ligand exchange precursor chemisorption is the Al_2_O_3_ ALD from TMA and H_2_O, the pathway of which is shown in Figure 4. During the first half-cycle of the reaction, TMA molecules chemisorb on a hydroxylated surface and react with the OH groups, where a ligand exchange occurs, and methane gas is released. After the surface is thoroughly saturated with the TMA molecules, all unreacted precursors and the by-products of the reactions are purged out. Water molecules enter the reactor in the second half-reaction and react with the CH_3_-terminated surface sites. Again, the ligand exchange happens between the CH_3_ groups of chemisorbed TMA molecules, and OH groups of water vapor and methane gas are released until all the active sites are filled. Lastly, by-products and unreacted water molecules are pumped out of the reactor [62]. The surface is now hydroxylated again, and the steps are repeated. Another example of ligand-exchange ALD reactions is the deposition of metal oxides using alkoxide precursors [74]. ALD of metals such as copper, ruthenium, and platinum often occurs based on dissociation chemisorption [72,75]. The association mechanism is the hardest to identify since it usually happens after a gas-phase dissociation or before a ligand-exchange reaction. A common example is ALD from metal halide precursors [72].

## 3. Theoretical Methods

### 3.1. Density Functional Theory

Density Functional Theory (DFT) is one of the computational methods based on the interaction between particles. DFT is used to investigate the electronic structure of many-body systems by reducing a 3N-dimensional problem to N 3-dimensional problems. In DFT, the structure of molecules may be predicted via the calculation of total energies and forces [75]. There are several examples where DFT has been broadly used along with ALD. DFT can be used to perform precursor design and comparison, predict ALD deposition characteristics such as overall growth, predict activation barriers and transition states, and determine reaction mechanisms such as initial surface reactions, reaction pathways, and the precursor chemisorption process [48,76,77,78,79,80].

### 3.2. Microscopic or Atomic Modeling Scale: Molecular Dynamics 

Molecular Dynamics (MD) can be used to simulate the interaction between particles [81]. In MD simulations, Newton’s equation of motion is integrated and numerically solved for simulating the movement of atoms and molecules [5,82]. MD simulation is capable of studying rather large systems for a relatively long time and results in the trajectory of the particles as a function of time [81]. For instance, when it comes to predicting the reaction pathways or precursor chemisorption processes in ALD, MD simulation is another method that is extremely helpful [75,83].

### 3.3. Lattice Boltzmann Method

The Lattice Boltzmann Method (LBM) is one of the methods used in Computational Fluid Dynamics (CFD), which was introduced in the late 1980s and is used for fluid simulations [84,85]. LBM is categorized as being on the mesoscopic modeling scale, which is appropriate for less complex systems as it needs less memory and has a short processing time [86]. This method employs the Boltzmann equation to model a fluid consisting of fictitious particles that are propagating and colliding [87]. LBM can be used along with ALD to simulate the flow of gases [86].

### 3.4. Off-Lattice Pseudo-Particle Method: Monte Carlo

Monte Carlo simulation is a stochastic computational technique to predict the probability of outcomes of various processes and obtain numerical results [88]. Monte Carlo can be used to solve intractable analytical problems or substitute for time-consuming or expensive experiments. Furthermore, researchers can employ this simulation technique to explore different aspects or modify the conditions of an experiment [88]. Random-based has a broad range of applications in finance, engineering, and science [89,90,91,92]. Monte Carlo simulation has been extensively used along with ALD for different purposes, e.g., determining film and precursor properties and the evolution of film morphology, modeling film growth, and studying the kinetics of reactions and the mechanism of materials processing in ALD [93,94,95,96,97]. Moreover, Kinetic Monte Carlo (kMC) simulation, which accounts for changes in the process with time, is invaluable in bridging the gap between individual reaction data from DFT and average growth characteristics from experiments [97].

### 3.5. Group Contribution Method 

The Group Contribution Method (GCM) can be employed to estimate the binary interaction parameters between different groups of atoms [98] where there is no available experimental data and can avoid the need for expensive experiments. In the GCM, the thermodynamic properties of a compound are predicted from its molecular structure. For this purpose, the molecule is split up into structural and functional groups composed of individual atoms or small groups of atoms. The GCM parameter of a functional group is estimated by the number of times a particular group appears on the adsorbent, multiplied by its contribution [99]. The GCM can be employed to estimate the thermodynamic properties of precursors used in ALD. For instance, ALD precursor molecules can be divided into smaller groups of atoms, and the GCM estimates the binary interaction parameters between those groups. Then, the activity coefficients of each group can be calculated based on those interaction parameters. Our group recently reported the thermodynamic properties of some well-known precursor molecules, which will be discussed in detail later [100,101].

### 3.6. Computer-Aided Molecular Design

Computer-Aided Molecular Design (CAMD) generates molecules with desired properties from functional groups using a reverse technique to that of the GCM. While the GCM estimates the molecular properties based on the functional groups comprising the molecule, CAMD, on the other hand, combines different functional groups to generate molecules having desired properties [102]. CAMD methods have been applied extensively in various areas such as extraction solvents [103,104,105,106,107,108], polymer designs [109,110], degreasing solvents [111], blanket wash solvents [112,113], absorption solvents [114,115,116,117,118], refrigerant design [119,120], distillation solvents [102,106,121,122], reaction solvents [123,124], catalysts [124], value-added products [125], crystallization solvents [126], and foaming agents [127]. CAMD has been used effectively to design novel clay-based adsorbents to adsorb radioactive elements from flowback/produced water [128] and remove arsenic from water [129]. Mukherjee et al. [110] used CAMD to design a novel polymer resin for metal ion removal from water. CAMD is also an appropriate method to be combined with ALD, especially to design novel precursor materials, which will be discussed later on [100].

## 4. Theoretical Studies on ALD

In the previous sections, different aspects of ALD as well as theoretical methods to study the ALD technique were briefly discussed. In this section, the previous research studies on theoretical ALD are reviewed. Table 2 summarizes the theoretical-study-only ALD articles found in the Web of Science and covered in this review. It is worth mentioning that this review mostly focuses on the theoretical-study-only ALD articles and does not cover the combined experimental and theoretical ones, which are cited here [75,76,130,131,132,133,134,135,136,137,138,139,140,141,142,143,144,145,146,147,148,149,150,151,152,153,154,155,156]. Figure 5 displays the publications, summarized in Table 2, that are covered in this study. Figure 5a illustrates the time evolution of publications on ALD studied with at least one of the theoretical methods mentioned above. In this figure, the theoretical-study-only articles are separated from the total set of articles, i.e., those that combine theory and experiments. During the last six years, researchers have focused more on combining experimental ALD studies with theory rather than performing a purely theoretical study. The reason most likely lies within the greater desire to validate experimental data with a computational model to establish their ALD method better. Figure 5b categorizes the publications based on the regions those studies were performed in.

Figure 6a presents the percentage of articles on ALD thin film materials studied with theoretical methods. Most researchers have focused on Al_2_O_3_ thin films presumably due to the simple and well-understood reaction mechanism of TMA and water. In this figure, the “others” section includes the theoretically studied ALD materials of yttrium oxide, silicon oxide, and ruthenium. Figure 6b–d visually categorize the research articles based on theoretical methods and ALD aspects (i.e., precursors, deposition characterization, and reaction mechanisms, respectively).

### 4.1. Precursors

First-principles theoretical methods can be used to select and optimize ALD precursors given any conditions. For instance, no experimental condition for ALD of silicon carbide (SiC) was known when Filatova et al. [178] performed DFT calculations to introduce the most promising precursors for this ALD system. They predicted that the combinations of disilane (Si_2_H_6_), silane (SiH_4_), or monochlorosilane (SiH_3_Cl) with ethyne (C_2_H_2_), carbon tetrachloride (CCl_4_), or trichloromethane (CHCl_3_) would be the most promising materials for ALD of SiC at 400 °C. As of then, SiC was only deposited via high-temperature CVD, and that was how they validated their method [178]. Therefore, those materials and conditions for ALD were proposed without performing a single experiment, saving a huge amount of time and energy. Thus, theoretical techniques can be used in a stand-alone manner without experiments, making them very powerful methods.

In 2014, Jung et al. [145] studied a newly synthesized zirconium precursor for ZrO_2_ ALD on silicon and compared their results with a commonly used zirconium precursor. Aside from exploring the properties of the precursor and film growth, they performed DFT to examine the initial growth mechanism of ZrO_2_ on hydroxylated silicon. According to their DFT calculations, the new precursor would result in a lower growth rate due to steric hindrance but, at the same time, a more distinct fraction of a monolayer during one cycle [145].

Dey and Elliott [48] introduced a copper (I) carbene hydride using DFT calculations that acted both as a reducing agent and a precursor for Cu ALD. They proposed a Cu-based reducing agent in case a co-deposition happened, which would be desirable as copper was still deposited [48]. In addition to Cu ALD [48,75,174], other materials have been studied through DFT, e.g., platinum [148], aluminum oxide [76,146,147], hafnium oxide [74,166,167], titanium oxide [77,169], and zirconium oxide [80,171]. Another study used DFT to design an aluminum ALD precursor by substituting one methyl in TMA and selectively decorating Pt nanoparticles by AlO_x_ via ALD [76]. Yang et al. [76] showed that dimethylaluminum isopropoxide (DMAI) could be used as an ALD precursor, and they predicted the decomposition mechanism of DMAI with DFT [76]. Recently, Park et al. [166] compared two different hafnium precursors for ALD. DFT was used to predict that the cyclic precursor would result in a lower growth rate compared with the alkylamide precursor due to the low probability of final chemical adsorption of the bulky cyclic ligand on the surface [166].

When a decision needs to be made between two similar precursors, DFT can be used to compare different ALD precursors without performing any experiments. For instance, halide precursors have been compared with regard to thermodynamics and kinetics [169]. Hu and Turner [169] compared TiI_4_ and TiCl_4_ from different aspects. Their results revealed that the difference in bond strength between Ti-I and Ti-Cl would not lead to a considerable change in the kinetics of their reactions with water. In contrast, different bond strengths significantly affected the reaction thermodynamics [169]. Without performing a single experiment, one would choose TiI_4_ over TiCl_4_ when a film with less impurity and a lower ALD temperature is more favored; for instance, TiI_4_ would probably be a better precursor for TiO_2_ ALD on organic substrates compared with TiCl_4_.

Recently, Shahmohammadi et al. [100] developed a theoretical method to design novel precursor materials for ALD. First, they developed a GCM model to predict the thermodynamic properties (i.e., activity coefficients) of the functional groups of already-existing ALD precursor materials such as tetrakis(dimethylamido)titanium (TDMAT), tetrakis (diethylamino) titanium (TDEAT), tetrakis (diethylamino) hafnium (TDEAH), and tetrakis (ethylmethylamino) hafnium (TEMAH). Then, using the estimated activity coefficients of those groups, they formulated a CAMD framework to optimally design novel precursor materials for ALD. Compared with the commercially available precursors, the most optimal designed precursors were predicted to have a ~40% increase in the ALD growth rate [100]. The same GCM model was employed to quantify the water impurity in an ALD reactor, which will be discussed in the next section [101].

### 4.2. Deposition Characterization

Different theoretical models can be used to describe the growth rate in ALD. In 2003, Puurunen [66] derived a mathematical model based on mass balance to describe ALD growth rate as a function of growth chemistry. The chemistry of growth was defined as the size of the reactants and their chemisorption mechanism on the substrate. Using that model, if the size of the ligands is known, one can simulate the growth per cycle from any compound with a certain chemisorption mechanism. The chemisorption processes of ligand exchange, association, and dissociation were theoretically described for the adsorption of the precursor on the substrate. It was found that either a steric hindrance or a limited number of active sites on the substrate causes surface saturation. According to the author, the steric hindrance of ligands would lead to the growth rate being less than a monolayer per cycle [66]. In another article, the same model was applied to three ALD systems, i.e., Al_2_O_3_ from TMA and H_2_O, Yttrium oxide (Y_2_O_3_) from Y(thd)_3_ and O_3_ (thd = 2,2,6,6-tetramethyl-3,5-heptanedionate), and TiO_2_ from TiCl_4_ and H_2_O. Based on that report, the ligand exchange would occur with all OH groups in the TMA/H_2_O process and with one or more OH groups in the Y(thd)_3_/O_3_ and TiCl_4_/H_2_O processes. Moreover, in all three systems, the steric hindrance of ligands likely defined the saturation [161]. A year later, the same author theoretically described random deposition as an ALD growth mode [56]. According to that research, random deposition and the growth rate value related the surface coverage to the surface fractions and the coverage of different layers. The surface fraction was defined as the difference between coverages of a layer and the previous layer, whereas the sum of those fractions resulted in the surface coverage [56].

Murray and Elliott [180] performed a comprehensive DFT investigation on ALD from 17 different metal precursors. Using hydrolysis models, they were able to predict the reactivity and stoichiometry of metal cation ratios in a variety of ternary oxides grown by ALD [180]. In another article, Elliot [78] developed a theoretical framework based on DFT to predict the growth rate and the intermediates of the reaction as well as their concentration in an Al_2_O_3_ ALD system. The well-known TMA and water system was employed for that study. The DFT analysis was performed with respect to the surface coverage of the intermediates, such as CH_3_ and H, and the effect of temperature as a processing condition was studied. According to the author, the higher processing temperature caused a higher concentration of hydrogen intermediates, reducing the growth rate. In addition, the lower temperature limited the mobility of hydrogen atoms, which decreased the growth rate in the system of that study [78].

In another TMA/water study by Puurunen [73], the growth rate was shown to quantitatively correlate to the concentration of hydroxyl groups on the surface before the TMA reaction. Based on the results of that study, the correlation parameters depended on the precursor chemisorption and steric hindrance [73]. The temperature dependency of the growth rate was also discussed in Puurunen’s study, where the temperature effect was related to the concentration of hydroxyl groups on the surface. As the substrate temperature increased, the concentration of surface hydroxyl groups decreased, which in turn reduced the growth rate [73]. The finding was in agreement with the previously discussed study by Elliot [78]. Both of those studies considered Al_2_O_3_ ALD using TMA and water, while, recently, Seo et al. [163] employed DFT to study the effect of different oxidizers (i.e., H_2_O_2_ and O_3_) reacting with TMA. According to their results, the molecular reactivity toward ALD of Al_2_O_3_ on a CH-terminated surface at low deposition temperatures for the three oxidizing agents was expected to be H_2_O < H_2_O_2_ < O_3_. Moreover, one H_2_O molecule was required to complete the reaction, whereas this number for both O_3_ and H_2_O_2_ was 2 [163]. DFT was used in another research study to investigate ALD-grown copper oxide nanoclusters on a porphyrin, supporting their similar reactivity characteristics with Cu-exchanged zeolites toward direct methane-to-methanol oxidation in a stepped conversion process [175].

One of the well-established numerical techniques extensively used in different areas is Monte Carlo simulation. Monte Carlo simulation was previously performed to explain ALD film growth in nanopores [96]. In that study, uniform film growth was observed up to 75 cycles, beyond which the film grew non-uniformly. At higher cycles, a thicker film was observed at the pore edge compared with the pore center, which was attributed to increasing the resistance to precursor diffusion at the pore mouth and the precursor depletion in the central regions with increasing the cycle numbers [96]. In 2018, Weckman et al. [97] performed a comprehensive study on the overall growth and surface chemistry of ZnO ALD by implementing a DFT calculation into a kMC model. According to the authors, the temperature dependency of film growth was in agreement with experimental data, while the growth rate per cycle was overestimated with simulation. Based on their report, the film growth in the reported system was low at lower temperatures, which was attributed to the high activation energy required to eliminate ethyl ligands of the precursor, which in turn caused impurities in the film. However, those barriers would be overcome at higher temperatures, resulting in a higher growth rate [97].

Moreover, the gas flow and temperature profile in ALD have been studied experimentally and theoretically. Burgess et al. [162] used CFD codes to simulate the gas flow and temperature profile in the AL_2_O_3_ and HfO_2_ ALD systems. They employed quantum calculations to identify reaction pathways and energies in both systems [162]. The gas flow and surface reactions in Al_2_O_3_ ALD using TMA and water were studied using 3D transient numerical models. The predicted growth rate under different ALD conditions (i.e., temperature and precursor purging time) was compared with experimental values. Slightly higher growth rates were predicted compared with the experimental results. The longer purge times at lower temperatures would result in closer agreement between the experimental and predicted growth rates. One interesting observation of that work was that the growth rate was dependent on the location of the sample in their ALD reactor, and the samples near the inlet of the reactor had a higher growth rate than those near the outlet [149]. Although their results showed good agreement between experiments and simulations, they only considered one full ALD cycle in their numerical study. A previous study suggested that after a specific number of ALD cycles where a rather thick film is formed, the growth mechanism and the growth rate may change since the effect of the substrate is diminished [6]. Pan et al. [86] compared two common LBM models to characterize the carrier gas flow in ALD: the lattice Bhatnagar–Gross–Krook (LBGK) and two-relaxation-time (TRT) models. According to their results, the TRT model was more reliable regarding stability, while the LBGK model was better in terms of computational efficiency [86]. 

Monte Carlo simulation is also applicable in implementing the kinetics of ALD. For example, Deminsky et al. [95] performed kMC simulation to study the temperature dependency of the growth rate and reported its slight decrease between 200 and 600 °C in Zr(Hf)O_2_ ALD. The observed temperature dependency of the growth rate was reported to be in agreement with experimental data and due to the steric repulsion between chemisorbed groups and adsorbed precursor molecules [95]. This is in line with another work, where the researchers used CAMD to design novel precursor materials for ALD, and the predicted growth rates displayed temperature dependency [100].

The morphology evolution of the films produced by ALD can be modeled and studied through Monte Carlo simulation. In a previous study, amorphous films deposited by ALD were examined with Monte Carlo simulation [94]. Based on the results, steric hindrance would delay the linear growth [94]. In addition, 3D Monte Carlo simulation has been used to determine the film thickness and sticking coefficients of TMA and bis-diethyl aminosilane (BDEAS) precursors in high-aspect-ratio 3D substrates [93]. In another work, Cremers et al. [179] reported the results of 3D Monte Carlo simulation in ALD on different large geometries, i.e., pillars versus holes. According to the authors, much less precursor exposure is needed for conformal ALD on pillars than holes, which, as a result, makes arrays of pillars more appropriate for ALD [179].

A recent report from the authors employed the GCM and ASST to quantify the impurity in an ALD reactor [101] reported earlier. When PMMA and a silicon reference were present in the same ALD reactor, the TiO_2_ growth rate on silicon was higher than stand-alone silicon. This elevated growth rate had been observed experimentally and was attributed to the released water molecules from PMMA pores [6]. With the help of the theoretical methods, the precise amount of water vapor from PMMA was calculated and confirmed the previously reported hypothesis. The predicted growth rate on the silicon reference was in agreement with the experimental data. The model also showed that the moles of water molecules acting as the second co-reactant of ALD decreased with increasing the number of ALD cycles [101]. This decreasing trend had been observed with experiments as well [6].

### 4.3. Mechanisms

The reaction mechanism happening in the Al_2_O_3_ ALD system using TMA and water has been studied theoretically to determine thermodynamically favorable pathways at various stages of ALD [157,158,159,164,165]. DFT cluster models have been used to predict reaction energetics and transition state structures for adsorption and a single ligand exchange during both the TMA and water half-reactions. Both half-cycle reactions are reported to be exothermic, happening through the formation of an Al–O Lewis acid–base complex followed by CH_4_ formation [164]. Another article was focused on atomic-scale models to investigate the reaction steps involved in the growth of Al_2_O_3_ ALD, especially precursor adsorption and by-product elimination [165]. Recently, reaction mechanisms between TMA and O_3_ were investigated using DFT. Some plausible intermediates of those reactions were found to be methoxy, formate, bicarbonate, and hydroxyl intermediates [160].

The dissociation of water molecules, when adsorbed on the substrate, hydroxylates the surface. Thus, when studying reaction mechanisms, the OH groups should be considered. DFT is used to study in detail energetics of the initial reaction pathways on the hydroxylated surface during the TMA pulse. TMA is reported to adsorb exothermically on the surface, reacting with the hydroxylated Al_2_O_3_ surface through ligand exchange reactions [157]. Travis and Adomaitis [158] used existing energetics data to determine kinetic parameters using statistical thermodynamics and absolute reaction rate theory. They presented a surface reaction kinetics and film growth model for the TMA half-reactions of Al_2_O_3_ ALD on a range of surfaces from bare to full hydroxylated states. According to their results, the reaction pathway differed if the surface was bare or hydroxylated, and maximum ALD growth only occurred at an initially saturating hydroxyl group density [158]. Brown et al. [83] performed MD on Al_2_O_3_ ALD, focusing on dissociation reactions. Based on their results, the studied variables, such as growth rate and surface roughness, were in close agreement with the experimentally reported values. However, the previous studies that only considered the ligand exchange reactions achieved less agreement with the experimental data [181].

Metal alkoxides were reported to be used as both hafnium and oxygen precursors in ALD. Mui and Musgrave [74] predicted the chemical mechanism of HfO_2_ ALD using DFT calculations, in which they examined Tetraethoxyl Hafnium (Hf(OEt)_4_) as both a precursor and an oxidizer. They studied different reaction pathways that would affect the ALD growth rate. According to the authors, incomplete surface elimination most likely led to carbon contamination and competed kinetically with the ligand exchange half-reactions. Additionally, the use of Hf(OEt)_4_ as the oxygen precursor was not advantageous as it violated the self-limiting characteristic of ALD [74]. The halide hafnium precursor was used in another work to examine the mechanisms of half-reactions happening in HfO_2_ ALD. It was reported that the adsorption energy and the preferred adsorption sites for metal precursors depended on the water coverage. That is, increasing the water coverage would lead to higher interaction between the metal precursor and multiple surface adsorption sites [167]. Another study reported the effect of cluster size on the formation of the HfCl_4_ complex during the precursor half-reaction [70].

The densification process in HfO_2_ ALD on SiO_2_ was studied through combined DFT and kMC [168]. The introduced possible densification reactions, which impacted growth evolution, were nucleation, the “inter-side” reaction, and the “on-site” reaction. The authors compared the simulation results of surface coverage to the experimental data up to 10 ALD cycles. They concluded that considering the densification reactions was the only condition that caused the agreement with experiments [168]. A year later, another group used DFT to study the reactions happening in HfO_2_ ALD [79]. They reported a detailed mechanism for that ALD system from tetrakis(dimethylamido)hafnium and water. Based on calculated activation energies, multiple proton diffusions from the surface to amide ligands and rotation of the protonated amine are more energetically favorable than ligand elimination in the initial stage. Thus, multiple proton diffusions to the amide ligands happen before the protonated amine ligands desorb. When the precursor was adsorbed on the surface, multiple protons diffused to amine ligands. That freed up hafnium to bond with oxygen, which had already been freed due to protonation and desorption of ligands and become five-coordinated. Then, hafnium became four-coordinated upon further ligand elimination. Hafnium became more highly coordinated as more ligands were eliminated, forming bonds with more oxygens and causing densification. The remaining ligands exchanged with hydroxyl groups during the water pulse, preparing the surface for the next precursor pulse [79].

Hu and Turner [77] employed DFT to investigate, in detail, the initial surface reactions of TiO_2_ ALD from TiCl_4_ and water. They studied different reactive groups of the SiO_2_ surface as the substrate, i.e., isolated hydroxyl groups, adjacent hydrogen-bonded hydroxyl groups, and surface oxygen bridges. According to their report, all the investigated surfaces were reactive toward TiCl_4_, where different intermediate species with different activation barriers were formed on each surface [77]. Later on, DFT was employed to compare the initial growth mechanisms in ALD of TiO_2_ and ZrO_2_ on SiOH surfaces using cycloheptatrienyl (CHT)–cyclopentadienyl (Cp) precursors [170,172]. According to the authors’ DFT calculations, the reactions happened through similar pathways in both metal oxides, where one hydrogen atom from the surface was transferred to the precursor ligand (Figure 7, reprinted with permission from reference [170]). However, the Zr precursor adsorption was exothermic and thus energetically favorable, whereas that of the Ti precursor was endothermic and thermodynamically unfavorable. The authors reported this difference to be a reason why an ALD window had been experimentally observed for ZrO_2_ growth and not for TiO_2_ growth [170]. The same researchers proposed initial reaction pathways for an ansa-metallocene Zr precursor ((Cp_2_CMe_2_)ZrMe_2_), where the steric hindrance of the large ligand prevented the reaction from occurring through chemisorption, and it happened due to the dispersion effect of the bulky ligand [173].

Han et al. [171] studied the initial growth mechanisms in ZrO_2_ ALD on hydroxylated SiO_2_ substrates by DFT. The intermediate complexes formed after each half-reaction of ZrCl_4_ and water were similar to the ones shown in Table 1. Furthermore, they reported that increasing the temperature would increase the precursor desorption and sub-monolayer growth. Another research study was performed on a similar system. Cui and Ren [80] investigated fourteen possible pathways of chlorine loss reactions in ZrO_2_ ALD using ZrCl_4_ and water through DFT. They reported that the HCl by-product did not prefer to self-eliminate throughout the process. Thus, eliminating chloride contaminations should be considered in that ALD system. Moreover, they discussed temperature effects on chlorine loss reactions, where the self-elimination of HCl was the dominant pathway at a lower temperature. At elevated temperatures, although the self-elimination of HCl was still favorable, hydrolysis was the dominant pathway due to the decrease in hydroxyl groups [80].

In addition to metal oxides, the mechanisms happening in metal ALD have also been investigated through DFT. For instance, Karasulu et al. [148] investigated platinum ALD on graphene using DFT. As a result, they were able to minimize the experimental procedures for process development and revealed that graphene oxide is an effective seed layer to obtain a uniform continuous Pt thin film. Ab initio DFT MD simulations were also helpful for investigating the thermal stability of simulation models [148]. Another interesting study used DFT to compare the reactions of two precursors in ruthenium ALD on different ruthenium surfaces [177]. The authors’ comparison between reactions on various substrates proved that a surface with a high surface energy and a complex topology, i.e., a surface with more defective sites for the adsorption and reaction, was the best substrate for ALD. Moreover, the presence of hydrogen atoms on the surface would prevent precursor adsorption [177].

In 2015, Hu et al. [75] studied the ALD of Copper(II) acetylacetonate (Cu(acac)_2_) on Cu(110) substrates, and investigated initial surface reactions of the copper precursor through DFT calculations and reactive molecular dynamics (RMD) simulations. Based on their results, the dissociation of Cu(acac)_2_ easily happened on Cu atoms, while the acac ligand could dissociate on both copper and copper oxide surfaces. Their DFT calculations revealed the sequential dissociation and reduction of the Cu(acac)_2_ precursor on the copper substrate (Cu(acac)_2_-Cu(acac)-Cu), which was in line with the reaction pathways they observed through RMD. Their RMD simulations revealed that the copper-rich surfaces were more reactive toward the precursor’s decomposition. They also reported that the atomic hydrogen was more reactive toward the precursors with hydrocarbon ligands than ozone or water as co-reactants [75]. Two years later, the mechanisms of metal ALD chemistry on an atomic scale were investigated through DFT [174]. The work by Elliot et al. [174] supported the reduction of a Cu(acac)_2_ precursor on Cu substrates, which had been reported earlier [75]. Another DFT study investigated the transmetalation reactions as the mechanism for Cu ALD using diethylzinc (ZnEt_2_) as the reducing agent [176]. The most thermodynamically favorable gas-phase reactions in both the precursor and co-reagent pulses are presented in Figure 8 (reprinted from reference [176]). The letter L in Figure 8 refers to the ligands of the reagents, meaning the reactions were generalized for commonly used Cu precursors, irrespective of which was used, while copper dimethyl-2-propoxide (Cu(dmap)_2_) was reported to be the best one [176].

## 5. Summary, Insights, and Future Challenges

In this review, all theoretical methods that have been employed so far to study ALD are discussed. The fundamental and ab initio (first-principles) techniques such as DFT are used to predict atomic behavior at the micro-level. These methods provide detailed information on atomic interactions without depending on experiments since they do not require initial data to be obtained. However, they are time-consuming and can become very expensive. On the contrary, macro-level methods such as CAMD are much faster and more cost-efficient than the aforementioned ones. These methods predict the optimal chemical molecules based on information about building blocks (functional groups) rather than atoms and can be used for reverse engineering. However, the main drawback of these techniques is the necessity of experimental data. For instance, one would need the properties of groups of molecules to use CAMD, requiring experimental data. Thus, whenever there is a lack of experimental data, the ab initio methods are useful, but if the experimental data are available, it would be more efficient to use reverse engineering methods.

Most of the researchers have so far focused on varying the operating temperature to theoretically predict the growth, film properties, and reaction mechanisms in an ALD system. However, the temperature is not the only factor affecting ALD reactions. Other factors, such as the substrate, gas flow, reactor design, and adsorbate–adsorbate and adsorbate–adsorbent interactions, affect the predicted mechanisms happening in ALD reactions. Those factors may be varied through computational methods to better predict the behavior of different ALD processes. In addition, most of the theoretical research articles on ALD so far have focused on the processes on inorganic substrates. However, since more and more experimental ALD studies are being performed on organic substrates, such as polymers and biomaterials, the actual growth behavior and reaction mechanisms on these materials are largely unknown. Due to the nature of organic materials, it would be highly complicated and challenging to predict the growth behavior, thin film characteristics, and reaction mechanisms on those substrates. We hope to see such interesting studies from theoreticians in the future to better assist in the design of experiments in those fields.

## Figures and Tables

**Figure 1 nanomaterials-12-00831-f001:**
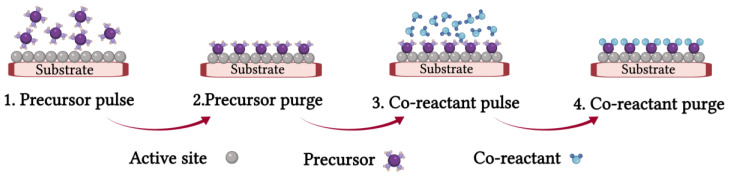
Schematic of each ALD cycle (Created with BioRender.com).

**Figure 2 nanomaterials-12-00831-f002:**
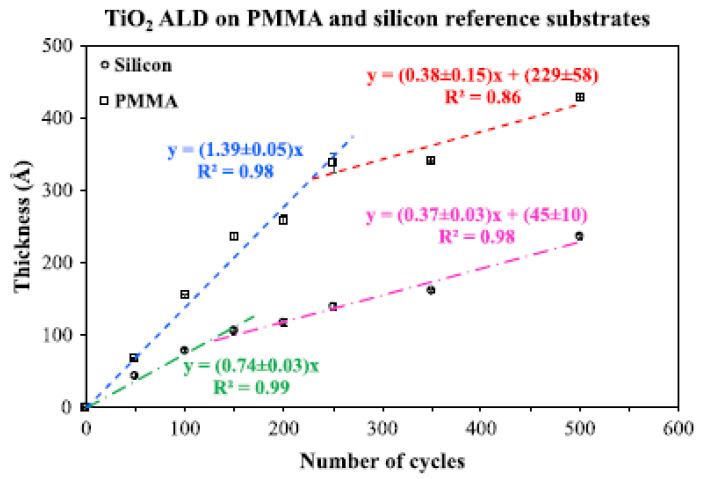
Duo-linear plot of TiO_2_ film thickness versus the number of cycles and a silicon reference (reprinted with permission from [6]).

**Figure 3 nanomaterials-12-00831-f003:**
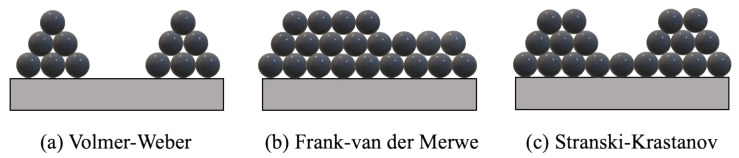
Schematic of common growth modes in ALD: (**a**) Volmer–Weber; (**b**) Frank–van der Merwe; and (**c**) Stranski–Krastanov.

**Figure 4 nanomaterials-12-00831-f004:**
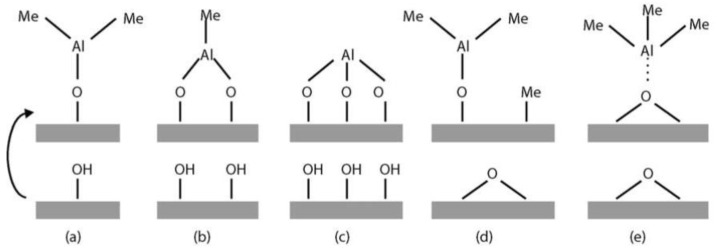
Schematic illustration of the TMA reaction possibilities on oxide surfaces: ligand exchange with (**a**) one, (**b**) two, and (**c**) three OH groups; (**d**) dissociation; and (**e**) association (reprinted with permission from [73]).

**Figure 5 nanomaterials-12-00831-f005:**
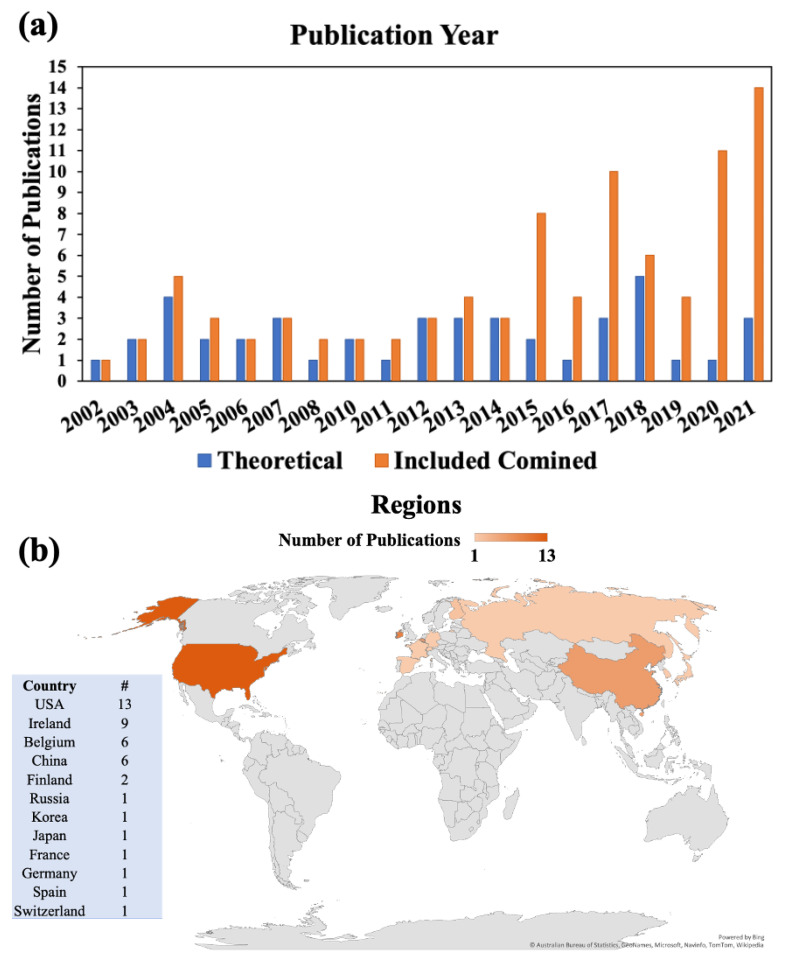
Illustration of the number of publications summarized in Table 2 categorized (**a**) per year between 2002 and 2021 and (**b**) per publication country. The search was performed with Web of Science using the following keywords: Atomic Layer Deposition and Theoretical. Irrelevant works were omitted and works combining experiments and theory are not covered in detail.

**Figure 6 nanomaterials-12-00831-f006:**
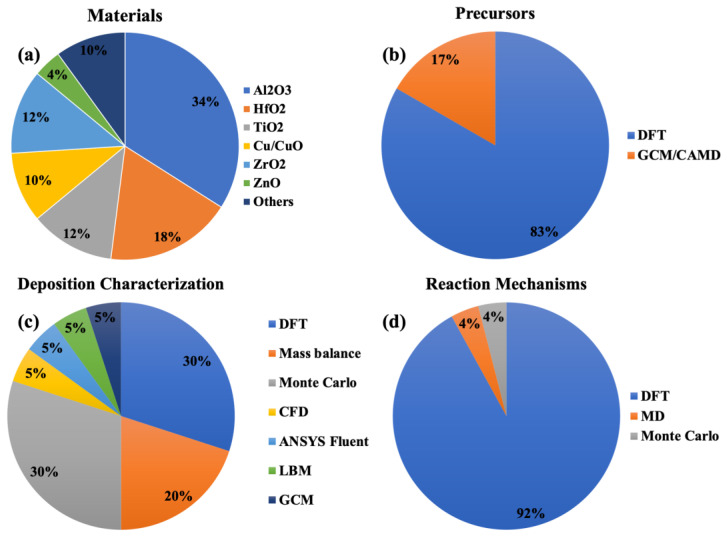
Theoretical studies of ALD categorized based on the theoretical method and (**a**) thin film material and the ALD aspect under study, i.e., (**b**) precursors, (**c**) deposition characterization, and (**d**) reaction mechanisms.

**Figure 7 nanomaterials-12-00831-f007:**
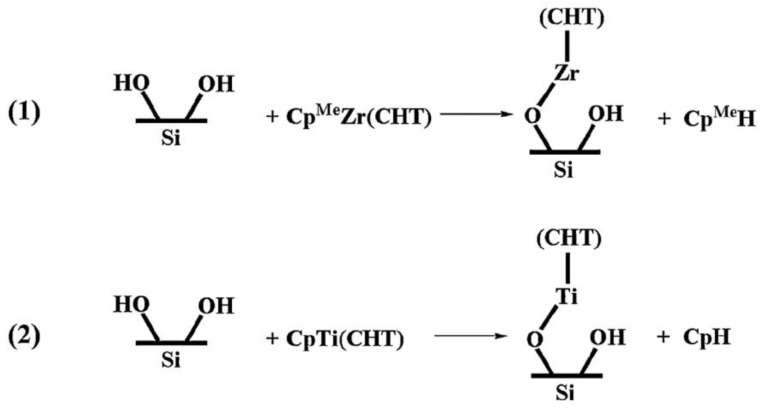
Surface reactions of Cp^Me^Zr(CHT) and CpTi(CHT) on the SiOH surface (reprinted with permission from reference [170]).

**Figure 8 nanomaterials-12-00831-f008:**
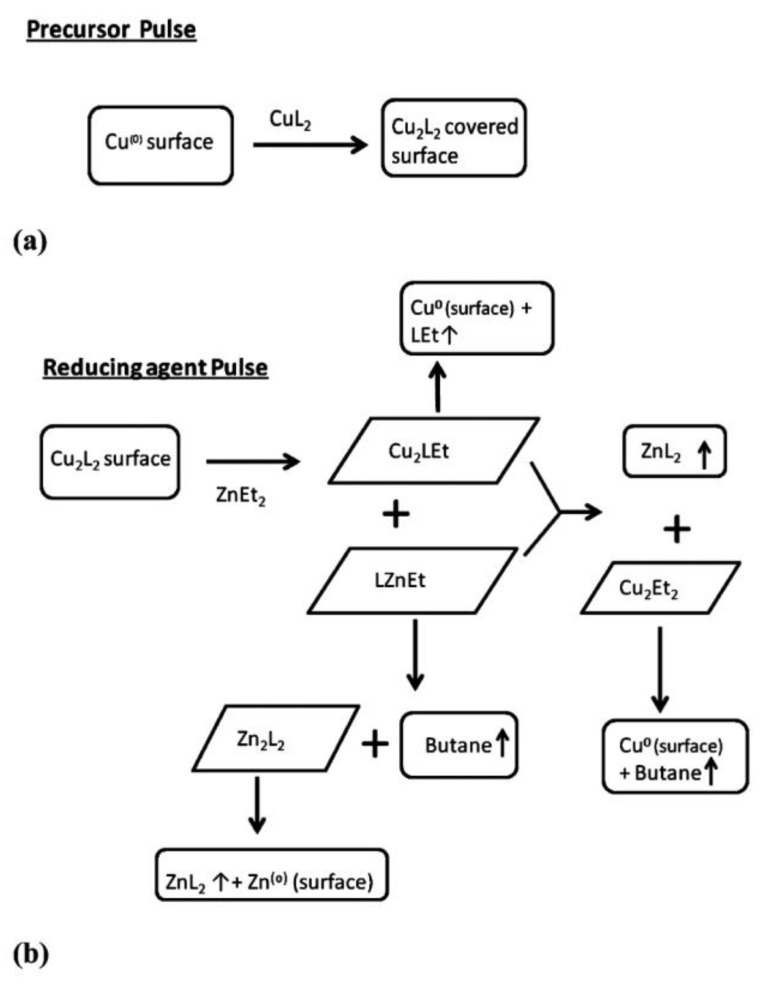
Flowcharts depicting possible ALD mechanisms for (**a**) a precursor pulse and (**b**) a reducing-agent pulse. Rectangular shapes denote starting reagents and end products, and slanted boxes denote intermediates. The upward arrows represent desorption of volatile species (reprinted with permission from [176]).

**Table 1 nanomaterials-12-00831-t001:** Well-known ALD reaction pathways.

Thin Film	Precursor	Co-Reactant	Reaction Pathway	Refs.
Al_2_O_3_	TMA ^a^	H_2_O	–OH + AlMe_3_→ –OAlMe_n_ + (3 − n) CH_4_ –AlMe + H_2_O → –AlOH + CH_4_	[62]
MO_2_ ^b^	MCl_4_	H_2_O	n (–OH) + MCl_4_ → (-O-)_n_MCl_4−n_ + n HCl (–O–)_n_MCl_4−n_ + (4 − n) H_2_O → (–O–)_n_M(OH)_4−n_ + (4 − n) HCl	[68,69,70]
MO_2_	TDMAM ^c^	H_2_O	M(NMe_2_)_4_ + 2 H_2_O → MO_2_ (solid) + 4 HNMe_2_	[71]

^a^ TMA, trimethylaluminum; ^b^ M = Ti, Hf, Zr; ^c^ TDMAM, tetrakis(dimethylamido)metal (Ti, Hf, or Zr).

**Table 2 nanomaterials-12-00831-t002:** Summary of the theoretical-study-only ALD articles.

Materials	Aspect of Study	Theoretical Method	References
Al_2_O_3_	Introduce new precursor Predict decomposition mechanism, chemisorption process, growth rate, intermediates of the reaction and their concentration, oxidizer reactivity, film thickness, and sticking coefficients Correlate growth rate quantitatively with hydroxyl group concentrations Simulate film uniformity, roughness, density, atomic ratio, gas flow, temperature profile, and surface reactions	DFT; Mass balance; Monte Carlo; CFD; Numerical model/ANSYS Fluent; MD	[73,78,83,93,96,157,158,159,160,161,162,163,164,165]
HfO_2_	Compare two precursors Predict growth rate and mechanisms Design novel precursors Simulate gas flow and temperature profile	DFT; GCM/CAMD; CFD; Monte Carlo	[70,74,79,100,162,166,167,168]
TiO_2_	Compare halide precursors Kinetics of reactions Design novel precursors Predict growth rate and mechanisms	DFT; GCM/CAMD	[77,100,101,161,169,170]
ZrO_2_	Predict mechanisms and growth	DFT	[80,170,171,172,173]
ZnO	Simulate growth rate and temperature dependency of growth	DFT/Monte Carlo	[97]
Zr(Hf)O_2_	Predict temperature dependency of growth rate (Kinetics)	Monte Carlo	[95]
Cu/CuO	Introduce new precursor Predict mechanisms and growth	DFT	[48,174,175,176]
Ru	Compare reactions of precursors	DFT	[177]
Y_2_O_3_	Predict chemisorption process	Mass balance	[161]
SiC	Introduce precursor	DFT	[178]
N/A	Simulate growth rate based on chemisorption process Describe growth mode Characterize carrier gas flow Model morphology evolution Compare precursor exposure on 3D substrates Predict cation ratios in ternary oxides	Mass balance; LBM Monte Carlo; DFT	[56,66,86,94,179,180]

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
