# Peer review of "Recent Advances in Theoretical Development of Thermal Atomic Layer Deposition: A Review"

_nanomaterials, 2022, doi:10.3390/nano12050831_

Round 1

Reviewer 1 Report

The paper by Shahmohammadi et al., is a review on existing publications dealing with theoretical aspects of the thermal atomic layer deposition processes. The paper is well written and covers large part (although, of course, not all) of the knowledge published to date. The paper also represents the topic quite attractively and can become of certain interest for readers.

One can notice, thet the authors have found and mentioned areas hardly covered so far, such as the deposition processes on organics. They conclude the following „...on organic substrates, such as polymers and biomaterials, the actual growth behavior and reaction mechanisms on these materials are largely unknown. Due to the nature of organic materials, it would be highly complicated and challenging to predict the growth behavior, thin films characteristics, and reaction mechanisms on those substrates.“

One may not completely agree that the treatment of the growth processes on organic materials would differ much from that represented by the material published so far. First of all, the main difference will then remain between the appropriate growth temperatures, since the organic substrates are much more temperature-sensitive compared to the inorganics. After that the issues worth addressing would remain related to the nucleation densities, which, in turn, should be quite similar on both inorganic and organic substrates at low temperatures, within the same precursor system.  

It is also somewhat „philosophical“ question about how much could theoretical calculations really predict the actual growth processes and to which extent they could become of help in planning and shortening the time of the experimental work. Nevertheless, criticism towards such kind of theoretical studies would be an exxaggeration and probably inadequate, as calculations are undoubtedly necessary before deeper cognition of the processes on surfaces.

The present manuscript can be published in the present form.

Author Response

We appreciate the comments from Reviewer #1. We agree with them on the provided comments about organic materials. Theoretical methods may ease the path to more comprehensive understanding of deposition on organic materials.

Reviewer 2 Report

The review article covers theoretical development of atomic layer deposition. The subject being dealt with is well specialized and quite interesting, but some of the description are incomplete and need to be revised. The authors have to revise the manuscript along these points mentioned below;

  1. # 240~ Increasing step coverage is still one of the technically difficult problems, and it varies depending on the type or variable of ALD process, so It needs to be modified to be ‘higher than conventional deposition processes such as CVD and PVD.’

  1. #531~ There is a conflict between the reference and this paper.

  1. #546~ No content cited in the reference can be found (ref. #105: growth rate slightly decreases with increasing deposition temperature).

  1. #613~ Many studies have reported that TDMA ligands are removed through beta hydride estimation with the lowest composition energy. It is necessary to confirm the relationship between proton diffusion and beta hydride estimation presented in the reference. Does it include each other? It is necessary to add a detailed description using Proton diffusion in the Hf-Opacking description.

  1. #670~ The experimental result of the reference is that the reactivity of H is larger than that of O3 in the case of hydrocarbon ligand such as acac, but the description in this paper seems to be written in a general situation. It should be limited to hydroncarbon ligand, or further investigation should be conducted on other types of ligands.

Author Response

Please see our responses to Reviewer #2 in the attached file.

Round 2

Reviewer 2 Report

Accept in present form